# Quantification of Lipids: Model, Reality, and Compromise

**DOI:** 10.3390/biom8040174

**Published:** 2018-12-14

**Authors:** Spiro Khoury, Cécile Canlet, Marlène Z. Lacroix, Olivier Berdeaux, Juliette Jouhet, Justine Bertrand-Michel

**Affiliations:** 1Centre des Sciences du Goût et de l’Alimentation, AgroSup Dijon, CNRS, INRA, Université Bourgogne Franche-Comté, 9E Boulevard Jeanne d’Arc, F-21000 Dijon, France; spiro.khoury@inra.fr (S.K.); olivier.berdeaux@dijon.inra.fr (O.B.); 2French LipidomYstes Network, 31000 Toulouse, France; juliette.jouhet@cea.fr; 3Toxalim, Research Centre in Food Toxicology, Université de Toulouse, INRA, ENVT, INP-Purpan, UPS, F-31027 Toulouse, France; cecile.canlet@inra.fr; 4Axiom Platform, MetaToul-MetaboHUB, National Infrastructure for Metabolomics and Fluxomics, F-31027 Toulouse, France; 5INTHERES, Université de Toulouse, INRA, ENVT, 31432 Toulouse, France; m.lacroix@envt.fr; 6Laboratoire de Physiologie Cellulaire et Végétale, Université Grenoble Alpes, CNRS, INRA, CEA, 38000 Grenoble, France; 7MetaToul-Lipidomic Core Facility, MetaboHUB, I2MC U1048, Inserm, 31432 Toulouse, France

**Keywords:** lipidomic, mass spectrometry, nuclear magnetic resonance, quantification, universal detector

## Abstract

Lipids are key molecules in various biological processes, thus their quantification is a crucial point in a lot of studies and should be taken into account in lipidomics development. This family is complex and presents a very large diversity of structures, so analyzing and quantifying all this diversity is a real challenge. In this review, the different techniques to analyze lipids will be presented: from nuclear magnetic resonance (NMR) to mass spectrometry (with and without chromatography) including universal detectors. First of all, the state of the art of quantification, with the definitions of terms and protocol standardization, will be presented with quantitative lipidomics in mind, and then technical considerations and limitations of analytical chemistry’s tools, such as NMR, mass spectrometry and universal detectors, will be discussed, particularly in terms of absolute quantification.

## 1. Introduction

Lipids represent a large and complex class of hydrophobic and amphipathic small molecules with a huge structural diversity (e.g., various combinations of fatty acyls and functional headgroups in phospholipids). They can be divided into eight basic groups according to the Lipid Mapsconsortium (https://www.lipidmaps.org/): fatty acyls, glycerolipids, glycerophospholipids, sphingolipids, sterol lipids, prenollipids, saccharolipids, and polyketides [1]. Changes in the level and/or in the composition of lipid species and/or classes occur after perturbation or during several physiological processes. Therefore, it is important to be able to profile the lipidome with a serious annotation [2] and to determine the absolute or relative abundance of one, several, or all lipids present in the sample of interest. However, the chemical heterogeneity of lipids, the occurrence of many isomeric and isobaric species and the large concentration range over which lipids are found, preclude the measurement of complete lipidome profiles with a single analytical method and greatly hampers the quantification of this family. Each lipid class (family), especially complex lipids, such as sphingolipids or glycerophospholipids, present a large number of molecular species, which complicate their absolute quantification. Fortunately, some processes are available to quantify these molecules but with some limitations. The aim of this review is to go through the different proposed techniques: universal detectors such Corona-CAD (charged aerosol detector) and ELSD (evaporative light scattering detector), NMR (nuclear magnetic resonance), and MS (mass spectrometry) with or without chromatographic separation, and to discuss their advantages and limitations for lipids quantification.

## 2. State of the Art for the Quantification of Lipids

Quantitative analysis of lipids has been rapidly expanding in the two last decades concomitantly with the advances in technologies such as MS [3] and NMR [4], and the number of published methods related to this is still growing. However, some cautions should be taken alongside the bioanalysis process to get an accurate result [5], especially when it comes to absolute quantification i.e., assigning an amount or concentration from the analytical response. Whatever the purpose of the study (-omics, toxicological, pharmaceutical, etc.), quantitative bioanalytical methods are critical for the interpretation of results, which underlines the need to have universal validation procedures. Thus, two workshops were held in 1990 and 2000 by regulatory agencies and scientific communities to harmonize validation procedures and to define the main parameters of validation with their criteria of acceptance [5]. Several guidelines resulted from these workshops, where methodologies and acceptance criteria are detailed for each validation parameter [6]. Indeed, they suggest using reference standards spiked in the same matrix of the intended studies to establish the calibration curve and quality control samples. These calibrators and quality control should be extracted with the same protocol as the studied samples and by using an internal standard, which should be the closest to the analytes. This last point is essential, especially when talking about absolute quantification. For lipids, we can distinguish two cases: the simple lipids; such as sterols and fatty acids with their derivatives, and the complex lipids; such as sphingolipids, glycerolipids, or glycerophospholipids. In the first case, pure analytical standards are available to prepare calibration curves. In the second case, hundreds of molecular species for each family can be detected for complex lipids in a biological sample. Today, 9856 species are listed on the Lipid Maps website (https://www.lipidmaps.org/) for glycerophospholipids and only about 80 analytical standards are commercially available. The same is true for sphingolipids: 4411 species can be found in the Lipid Maps website and very few species are available. Under these circumstances, we will not be able to obtain calibration curves for each molecular species, and as we know that detection is sensitive to the nature of the molecule (especially with fatty acids), the absolute quantification will not be possible. Furthermore, a full validation process should be performed prior to the assays, including the evaluation of selectivity, concentration range of the calibration curve, accuracy, precision, recovery, and sensitivity [7]. Moreover, it is recommended that assays are validated routinely by performing a new extracted calibration curve and/or a qualitative control sample series every day. These validation steps could be very demanding when several analytes, with different physico-chemical properties, should be analyzed in the same run, involving tedious sample preparation and extraction steps. Moreover, these guidelines are mainly addressed to pharmaceutical and toxicological applications implying that they are mostly established for the quantification of exogenous compounds in biological matrices. In the case of lipidomic studies, where compounds of interest are endogenous, absolute quantification can be achieved via a standard addition method [8], using surrogate matrices, or using isotopic dilution when lipids are detected by mass spectrometry [9]. However, lipid analytical standards are not commercially available in most cases meaning that only relative quantification, i.e., comparing the amount of the analyte to an analyte of reference, could be applied [10].

## 3. NMR Techniques Can Provide Absolute Quantification of Lipids

### 3.1. Mechanisms and Response of Nuclear Magnetic Resonance

Nuclear Magnetic Resonance spectroscopy is a well-known analytical technique for structure elucidation and quantification. The NMR principle is based on the existence of atomic nuclei in specific nuclear spin states when exposed to an external magnetic field. This technic observes transitions between these spin states that are specific to the particular nuclei in question, as well as that nuclei’s chemical environment. Only some isotopes, determined by their number of protons and neutrons, can give NMR signals, including ^1^H, ^13^C, and ^31^P. The two most important NMR-measured interactions are the chemical shift depending on the chemical environment of the nucleus and the couplings between nearby nuclear spins. These interactions give information about the chemical structure of the metabolite. NMR spectroscopy is also a good analytical technique for quantitative estimation.

The most important fundamental relationship of quantitative NMR is that the signal integration (*I*) in the 1-D NMR spectrum is proportional to the number of nuclei (*N*) responsible for that resonance:(1)I=K×N
where *K* is the spectrometer constant and remains the same for all resonances in an NMR spectrum. *N* is related to the concentration of the compound of interest *X* [*X*] using the following expression: (2)N=p×[X]×V
where *p* is the number of equivalent nuclei responsible for the resonance, and *V* is the volume of the sample. The concentration ratio [*A*]/[*B*] between two compounds *A* and *B* can be easily calculated by employing the following expression:(3)$$\frac{\left[A\right]}{\left[B\right]}=\frac{IA}{IB}\times \frac{pA}{pB}$$
Therefore, using a reference compound (internal or external) with a known concentration, it is possible to calculate the concentration of all compounds present in the sample. Several experimental parameters (e.g., relaxation delay, pulse sequence, and acquisition time) can affect the quantitative accuracy and precision, and these parameters have been discussed by Bharti et al. [11].

### 3.2. Applications to Lipid Analysis

^1^H and ^31^P NMR spectroscopy can be used for the quantification of lipid species because they are the most sensitive nuclei.

In proton NMR spectroscopy, each magnetically non-equivalent hydrogen nucleus in each lipid species, will exhibit an NMR signal at a characteristic resonance frequency, which is measured as a chemical shift relative to a standard compound. We know that the chemical shift depends on the chemical environment of the proton, and protons of similar molecules will give signals with close chemical shifts. Lipid species contain many long-chain fatty acids, so many signals of protons (e.g., (CH_2_)_n_ in fatty acids (FAs)) will be overlapped and the detailed characterization of lipid species is unfeasible. However, it is possible to identify different types of lipids such as cholesterol, esterified cholesterol, triacylglyceride, phospholipids, and unsaturated fatty acids. Very detailed assignments of the chemical shifts were summarized in the review by Vosgaard and Guo [4]. The quantification of signals is possible only for isolated peaks without signal overlapping. 

In most studies using proton NMR based lipidomics, relative quantification or spectral binning is performed followed by multivariate statistical analysis—such as principal component analysis—to highlight changes in lipid composition due to diseases, poison induction, or disorder [12,13].

Only few studies reported the use of absolute quantification of lipids by proton NMR spectroscopy because of overlapped signals. Some lipid classes (total cholesterol, total triacylglycerides, total phospholipids) can be easily quantified using the signal integration and an internal or external standard. Srivastava et al., determined the absolute concentration of triacylglycerides (TG), total phospholipids (PL) and total cholesterol in serum samples using quantitative proton NMR spectroscopy and an external standard [14]. The signals at 0.68 ppm, 3.95 ppm and 4.14 ppm were used for the quantification of total cholesterol, total PL, and total TG respectively. They showed that the concentration of triacylglycerides, phospholipids and total cholesterol was significantly higher in DMD (Duchenne muscular dystrophy) patients as compared to healthy subjects. To quantify more lipid species, Barrileo et al., developed a new bioinformatics tool for quantitative ^1^H NMR lipid profiling [15]. This tool, LipSpin is a graphical user interface software that allows the quantification of lipids in proton NMR spectra. Lipspin imports data samples from either raw free induction decays or one-dimensional NMR (1D) processed spectra and NMR signals were quantified with line shape fitting analysis. After the quantification process, signal areas can be converted into molar concentrations by using an internal standard of known concentration, or an external measurement, for example using total cholesterol concentrations determined with other methods. In this study, Barrilero et al. quantified in plasma samples total FAs, saturated fatty acid (SFA), ω-9 FA, ω-6,7 FA, ω-3 FA, mono-unsaturated fatty acid (MUFA), arachidonic acid and eicosapentaenoic, docosahexaenoic acid, Linoleic acids, free cholesterol, cholesterol ester, total cholesterol, triacylglycerides, phosphatidylcholine, sphingomyelin, lysophosphatidylcholine and total phospholipids. Results were compared with those obtained using gas chromatography (GC) coupled to FID and enzymatic methods.

Lipoprotein profiling of plasma samples without extraction, can be useful for the study of diseases [16]. A 1D ^1^H diffusion NMR experiment was recorded to detect large molecules (such as lipoproteins, glycoproteins and choline compounds) and lipoprotein profile characterization was performed from NMR data using Liposcale test [17]. In this tool, the methyl signal was surface fitted with nine Lorentzian functions associated with each lipoprotein subtype: large, medium and small of the main types of lipoprotein—very low density lipoprotein (VLDL), low density lipoprotein (LDL), intermediate density lipoprotein (IDL) and high density lipoprotein (HDL). The area of each Lorentzian function was related to the lipid concentration of each lipoprotein subtype, and the size of each subtype was calculated from their diffusion coefficient. The diffusion NMR data was also used to quantify choline compounds (3.3–3.18 ppm) and glycoproteins (2.15–1.9 ppm), based on peak deconvolutions. In this study, lipoprotein profiles showed significantly lower plasma concentrations of cholesterol-IDL, triacylglycerides-IDL and HDL in mothers of growth-restricted fetuses compared to controls. ^1^H NMR-based lipidomic analysis of HDL lipoprotein of plasma samples, isolated from non-HDL lipoproteins by precipitation, helps in the characterization of the atheroprotective function of HDL and the identification of novel biomarkers of cardiovascular risk [18]. Jimenez et al. reported an extensive NMR trial of quantitative lipoprotein in human blood serum and plasma. They quantified 105 lipoprotein subclasses without physical separation, using an algorithm based on the deconvolution of the lipid signals combined with multivariate statistical model building [19].

For quantification of phospholipids, ^31^P NMR spectroscopy is the most selective analytical tool. The ^31^P nucleus has a natural abundance of 100% which results in a high sensitivity and each PL has only one phosphorus with a characteristic chemical shift. Assignment of PL species can be confirmed using two dimensional ^31^P-^1^H NMR spectroscopy or by addition of each individual PL species in the sample. Using an internal standard, like triphenylphosphate with known concentration, it is possible to quantify the different classes of phospholipids. As discussed by Li et al [4], it is necessary to use a mixture of solvents (CDCl_3_, MeOD, D_2_O-EDTA) to obtain a well resolved spectrum. Bettjeman et al. determined the absolute concentration of phospholipids in five marine tissues using quantitative ^31^P NMR spectroscopy [20]. They quantified 17 PL species. Due to the low natural abundance of ^13^C, ^13^C NMR is not as sensitive as ^1^H and ^31^P, and no study reported absolute quantification of lipid species using ^13^C NMR spectroscopy.

Alternatively, two dimensional (2D) ^1^H NMR is an appealing solution in order to obtain a better resolution of the lipid signals, as it allows to split the signal in two dimensions. These experiments are difficult to apply to lipidomics, as they involve long experiments (several hours), and absolute quantification requires calibration procedures. Marchand et al used 2D UF COSY (Ultra-Fast COrrelation SpectroscopY) NMR experiment of serum samples of pigs exposed to ractopamine [21]. Relative quantification was performed in this study, but they showed the potential of fast multidimensional NMR experiment for lipids quantification since the analysis duration is low (26 min) and the peaks are well-resolved.

Contrary to MS, the detailed characterization of lipid species by proton NMR spectroscopy is unfeasible as many protons in FA with similar chemical environment give largely overlapped resonances and the quantity of material to acquire an accurate spectrum is 100 times higher than what is needed for mass spectrometry analysis. ^1^H NMR has some drawbacks such as low sensitivity, signal overlapping and discrimination of resonances. However, ^1^H NMR spectra of the lipids in biological matrices provide a fast overview of the major lipid classes (fatty acids, glycerolipids, phospholipids and sterols), and its spectral linearity avoids the use of multiple internal standards for quantitative estimation. ^31^P NMR is a powerful technique for the quantification of phospholipid classes. NMR is therefore a very promising tool for lipidomics, as illustrated by a recent perspective on the subject [4].

## 4. Universal detectors are good tools to quantify lipids

### 4.1. Chromatographic Separation and Detection Methods

Numerous chromatographic approaches can be applied to analyze lipids in biological matrices. Normal phase high-performance liquid chromatography (NP-HPLC) and hydrophilic interaction liquid chromatography (HILIC) are usually used for the separation of lipid classes, whereas reversed-phase high-performance liquid chromatography (RP-HPLC) is used for the separation of lipid molecular species. Supercritical fluid chromatography (SFC) can also be used in the field of lipid analysis [22]. When analyzing lipids by liquid chromatography (LC), lipid detection is usually performed by several techniques. As lipids do not usually present exploitable chromophores, they are rarely detected by classic methods like ultraviolet (UV). However, UV detection is the most common procedure used for targeted analyses of lipids containing a conjugated double bond system or aromatic rings such as carotenoids, tocopherol and tocotrienol [23], oxidized lipids [24] and conjugated fatty acids [25,26]. Mass spectrometry coupled to different ionization sources (electrospray ionization (ESI), atmospheric pressure chemical ionization (APCI) is also widely used to identify and quantify lipids. However, quantification results using this technique can be influenced by the ion suppression phenomenon [27]. To quantify lipids in biological matrices, we can use universal nebulizer detectors such as evaporative light scattering detectors (ELSD) or charged aerosol detectors (Corona-CAD®). The use of universal detectors avoids ion suppression effects but requires the correct separation of the lipid classes by chromatography.

### 4.2. Mechanism and Response of Universal Detectors

Charlesworth et al. described the principle of ELSD [28], whereas the most recent Corona-CAD was described by Dixon and Peterson [29]. The operating principle of these detectors is comprised of several steps. The principle is similar between the detectors in the first two steps of functioning, nebulization and evaporation and is different in the subsequent steps. After nebulization, the particles’ size (*D_sv_*) can be measured based on Nukivama and Tanasawa’s formula [30]:(4)$$D_{sv}=\frac{585\sqrt{\sigma }}{\left(u_{g}-u_{l}\right)\sqrt{P_{l}}}+597{\left[\frac{\mu }{\sqrt{\sigma_{l}P_{l}}}\right]}^{0.45}{\left[1000\frac{Q_{l}}{Q_{g}}\right]}^{1.5}$$
where *σ*, is the surface tension of liquid (dyne/cm); µ, liquid viscosity (poise); *ρ_l_*, liquid density (g/cm^3^); *Q_g_*, gas flow rate (cm^3^/s); *u_g_*, gas velocity (cm/s); *Q_l_*, mobile phase flow rate (cm^3^/sec); and *u_l_*, liquid velocity (cm/sec). This equation has two terms containing some factors that can influence the nebulization and therefore the particle size. In both terms, when parameters concerning *Q_g_* or *u_g_* are increased, a decrease in particle sizes is observed. In addition, the use of solvents with a high density can also decrease the size of these particles. Thereafter, the evaporation can decrease the particle size, which can be calculated using this formula:(5)$$d=D_{0}{\left(c/\rho_{s}\right)}^{1/3}$$
where *d* is the particle size after evaporation, *D*_0_ is the size of initial droplet, *c* is the concentration, and *ρ_s_* is the density of the analyte. Concerning Corona-CAD, the mechanism depends on charge transfer to the particles after collision with nitrogen that is already positively charged due to the Corona needle. Finally, the charged particles are detected by an electrometer. Generally, the attributed charge depends on the particle size. For ELSD, the response is distinguished by the diffusion of light by several phenomena (Rayleigh, Mie, and refraction–reflection). When arriving at the detection cell, the particles are crossed by a light beam. The amount of scattered light is measured by a photomultiplier. The light scattering depends on the size and the morphology of the particles, the angle of collection of the light, the surface, and the wavelength of the incident light. Both universal detectors have a nonlinear response, which can be described by the following equation:(6)$$Y=a\times m^{b}$$
where *Y* is the detector response and *m* is the mass of the analyte; *a* and *b* are parameters that depend on the analyte and chromatographic conditions. The response is independent of the chemical structure of the analyte like the degree of unsaturation and chain length. However, the detector signal depends on the mass of analytes [31]. Thus, these detectors are largely used for lipid quantification.

### 4.3. Applications to Lipid Analysis

Universal detectors are compatible with all solvents that are more volatile than the analyte [32], as well as with gradient systems, which are preferred to separate lipids covering a large-scale polarity. Christie et al., were the first to introduce the use of NP-HPLC coupled to ELSD, enabling the separation of lipid classes from animal tissue [33]. This separation was performed in about 20 min using silica as a stationary phase and a ternary elution gradient. Besides silica, diol columns were also used to separate PL classes from lecithin samples [34]. Technological developments led to the use of other stationary phases like polyvinyl alcohol (; PVA-Sil^®^), which represents a real benefit for lipid analysis using NP-HPLC coupled to universal detectors with complex gradients. Retention properties of PVA were evaluated by Deschamps et al. for lipid class analysis sphingolipids, glyceroglycolipids and phospholipids using ELSD with a binary elution gradient [35]. Godoy Ramos et al., evaluated the use of ELSD and CAD for PL class analysis in the membrane of Leishmania using NP-HPLC with a PVA column [36]. A ternary elution gradient was used to separate PL classes in 68 min. The response of both nebulizer detectors was not linear, and Corona-CAD was more sensitive and more precise than ELSD for the low mass ranges. A linear model for an injected mass of 30 ng to 20 µg can describe the response. The limit of detection (LOD) and the limit of quantification (LOQ) were three times lower with Corona than ELSD [36]. Mobile phases in the last two examples were supplemented with triethylamine (TEA) and formic/acetic acids, respectively. Generally, the addition of 0.1% (*v*/*v*) TEA and an equimolar amount of formic or acetic acids to the mobile phase improved the ELSD response and peak resolution [37]. Other published works using LC-ELSD have addressed the separation and the quantification of PL classes from different biological matrices, such as soybean [38], brain [39], egg yolk [40], plasma [41], etc., with a binary or ternary system of solvent mixtures.

Few applications have addressed the use of a HILIC system coupled to ELSD for lipid separation. Donato et al. showed quantitative analysis of five PL classes, after solid phase extraction (SPE) extraction, from milk samples in 55 min using HILIC-ELSD. The LOD and LOQ were between 1.6–4 µg/mL according to the analyzed PL classes [42]. ELSD was also used in RP-HPLC, as mentioned by Zhong et al. [43] for quantitative analysis of the compounds of cationic liposomes using a C18 HPLC column. Some cholesterol and phosphatidylcholine (PC) species and their degradation products were determined in liposome-based drug formulations with LOD and LOQ of 0.15 μg and 0.30 μg respectively [43]. Furthermore, the first application of SFC-ELSD to separate the major PL classes in soybean lecithin on packed columns was performed by Lafosse et al. [44]. In addition, the coupling of SFC with ELSD allowed for the separation of about 15 triterpenoids compounds from apple extracts with isocratic conditions [45]. However, the use of ELSD coupled to SFC, especially for quantitative applications, remains limited compared to MS because of the low sensitivity and the nonlinear response [46].

Regarding the use of the most recent detector Corona-CAD, a method in NP-HPLC was firstly developed for quantitative analysis of non-polar lipid classes (phytosterol esters, triacylglycerols, fatty acids, and free phytosterols) in vegetable oils with a LOD of 1 ng [47]. Other methods were developed in NP-HPLC-CAD to quantify polar lipid classes (glycolipids and PL) from vegetable extracts with 25 ng as LOD [48]. CAD was also coupled to a NP-HPLC system to separate the natural isomers of tocopherols and tocotrienols [49]. Although the CAD could be used to quantify some tocopherols and tocotrienols, the presence of other unknown peaks with the same retention time complicates the chromatogram. Thus, under such conditions, it seems that fluorescence detection is still the method of choice in LC for quantitative analysis of tocopherols and tocotrienols [50].

Coupling CAD to the LC system under HILIC conditions was mentioned by Le Bon et al. to separate and quantify PL classes in olfactory tissues of rats and mice [51]. Otherwise, a method in RP-HPLC using CAD as a detection method was developed to quantify triacylglycerides species (triolein and trilinoline), as well as other non-polar lipids in vegetable oils [52]. The baseline was noisier with the RP-HPLC system compared to NP-HPLC, and the LOD was around 25 ng. Nair and Werling addressed the use of LC-CAD in order to quantify free fatty acids resulting from the hydrolysis of PL in a pharmaceutical suspension formulated with PL as a stabilizing agent [53]. In this study, a Corona-CAD performance was compared with ELSD. Results showed a better sensitivity, linearity, fidelity, and recovery with Corona detection for the evaluated parameters.

As shown in the various applications above, the universal detectors are largely used for quantitative analysis of lipid classes and species. These examples showed the possibility of coupling these detectors with different methods of liquid chromatographic separation with more applications in NP-HPLC to quantify lipid classes and some limitations in SFC. Moreover, experiments conducted in NP-HPLC led to less noisy baseline, especially when using Corona detector. As the response of universals detectors is independent of chemical structure, coupling these detectors to RP-HPLC could be used to quantify lipid species without impacting the response by ion suppression effects [27]. Comparative studies evaluating the performance of these detectors demonstrated that the most recent detector (Corona-CAD) has a better sensitivity, precision and repeatability compared to ELSD. In addition, the response of CAD could be linear at low concentrations. Knowing that the response of both detectors is dependent on the particle size of analyzed lipid, parameters including gas pressure, attenuation, nebulizer gas flow rate, nebulization and evaporation temperatures can be optimized in order to improve the detection. Both detectors have the advantage of performing absolute quantification when lipid standards are available. Quantitative analyses can also be improved by the use of internal standards. The literature shows many other applications for lipid analysis using universal detectors. Finally, the choice of the detector and analytical methods in lipidomic studies seems to be relevant in relation to the physico-chemical nature of the molecules to be analyzed and according to the objective of the study.

## 5. And what about Mass Spectrometry?

Mass spectrometry is a powerful analytical technique used to quantify known materials, to identify unknown compounds within a sample, and to elucidate the structure and chemical properties of different molecules. The complete process involves the conversion of the sample into gaseous ions, with or without fragmentation, which are then characterized by their mass to charge ratios (*m*/*z*) and relative abundances. An MS instrument consists of three components: the ion source for producing gaseous ions from the studied substance, the analyzer for resolving the ions into their characteristic mass components according to their mass-to-charge ratio, and the detector system for detecting the ions and recording the relative abundance of each resolved ion specie. MS suppliers propose different ion sources, such as ESI [54,55,56] or APCI [57,58,59], which can be used for lipid analysis. The efficiency of ionization can be different depending on the family of lipids, and even more complicated according to the structure (chain length and number of unsaturations) of molecular species within the same family. This variation is linked to the geometry of the source, to ionization conditions (e.g., flow rate, ion source temperature, and ionization voltage) and to tuning conditions. This significant change in ion peak intensities depending on lipid species structure is a crucial point for the quantification. The analyzer can be of different natures, low resolution MS such as triple quadrupole (QqQ) and high resolution MS (HRMS) like TOF (Time of Flight) or Orbitrap. Triple quadrupole MS tuned to multiple reaction monitoring (MRM) mode not only facilitates the detection of lipids with very low abundance by virtue of its higher sensitivity, but also confers higher quality of MS data both in terms of specificity and quantitative accuracy with a large dynamic range [60]; it is well adapted to targeted lipidomic. High resolution MS can provide users with unrivaled resolution and the critical advantage of accurately defining the exact elemental composition of individual ion peaks over MS with unit resolution. These technologies had a lower linear range compared to QqQ, but this point is greatly improved with the new generation of these instruments. With MS detection, the calculation of individual lipid concentration from lipidomic datasets comprises different steps and requires specific assumptions that affect the estimation. These steps include (i) normalization with the corresponding ISTD; (ii) correction for isotopic overlap and isotope distribution effects; (iii) normalization for the starting sample amount; (iv) calculation of absolute concentrations, based on spiked ISTD, calibration curves and response/correction factors; and (v) drift/batch corrections [61]. This last point is particularly difficult to obtain especially for complex lipids.

### 5.1. Internal Standards and Quality Control Are Essential

Accurate quantification in lipidomics requires a method to control the variability of lipid extraction, ionization efficiency and systemic drift in the mass spectrometer. The addition of internal standards to samples allows compensation for these sources of variability [62]. However, the choice of internal standards is critical for accurate quantification. Ideally, internal standards should have the same chemical and physical properties as the lipids to be quantified, but distinguishable in mass spectra. Therefore, a stable-isotope labelled internal standard for each lipid specie is the preferred option for accurate quantification. Usually, lipidomic experiments measure several hundred of lipid species in a same run. Thus, it is impractical to obtain stable-isotope labelled variants for each lipid species. In practice, it is common to use one or two internal standards for each lipid class. Nonetheless, stable-isotope labelled internal standards are not often available for all lipid classes. Therefore, non-physiologically occurring lipid species (odd- or short-chain variants) can be used instead of them [5]. One way to obtain a complete panel of labeled standard is to grow cell on labeled substrate, that is easy to obtain with micro-organism such as *Escherichia coli*, the extract obtained from such a culture will be fully labeled and will be perfect to analyzed *E Coli* in metabolomics [63] but it will be unfit for eukaryotic analysis (lipids family are very different). And unfortunately, to obtain fully labeled yeast [64] or worse cell extract is still complicated, this extract will be a great progress for quantification of lipids. In addition to internal standards, quality controls should be in place to ensure that systematic variation does not occur during an experiment. Samples should also be randomized prior to sample preparation to avoid systematic variation from being misrepresented as group differences. However, we should admit that this point might be complicated for lipidomic analysis, due to the diversity of lipid families, it is not very easy to find reference material. National Institute of Standards and Technology Standard Reference Material 1950 (SRM 1950) “Metabolite in Frozen Plasma” [65,66,67] can be good alternative and can be extracted alongside test plasma samples to serve as a matrix-matched quality control for metabolomics and lipidomics applications. However, certified values for SRM 1950 are only available for selected metabolites (e.g., amino acids, vitamins, carotenoids, etc.), fatty acids, and total cholesterol. The comparison to this reference material can be easily obtained with LipidQC software [68]. This is a good option even if its stability is not guaranteed. In any case, the easiest way to have a quality control is to use a pool of all the samples to be analyzed, or if it is not convenient (e.g. for cell or tissues) a pool of the extract ready to analyzed, which can be regularly used during the experiment.

### 5.2. The Absolute Quantification of Simple Lipids

Quantification of simple lipids, such as fatty acids and their derivatives, sterols and their derivatives, vitamin, sphingoid base, is easy to perform. Primary standards are commercially available from different suppliers, and chemists are deeply engaged to develop complex total organic synthesis to provide these molecules [69,70,71]. Labeled molecules are not available for all the primary standards, but analytical chemist can at least synthesize few internal standards for each family. Even if blank matrix does not exist since lipids are endogenously present in most matrices, absolute quantification will be possible. Indeed, sample preparation could be validated with internal standards, and calibration curve could be run in solvent or in matrix after extraction. A lot of examples exist for absolute quantification of simple lipids like sterol by GC-MS [72] or by LC-MS [57], bile acids by GC-MS [73] or LC-MS [74] for fatty acid mainly by GC-MS [75], or their derivatives mainly by LC-MS [76,77,78] or for sphingoid base by LC-MS [79].

### 5.3. Can We Perform Absolute Quantification for Complex Lipids?

As previously shown, the lack of analytical standards for all lipid classes and molecular species compromises the absolute quantification of complex lipids. In the best cases, accurate or even close quantification can be achieved. Nevertheless, the situation is a little different depending on the use or not of a chromatographic separation before the detection by mass spectrometry.

#### 5.3.1. With Shotgun Lipidomics

Applying shotgun lipidomics process means that, the crude lipid extract is directly infused trough the electrospray ionization source. This technique is renowned for its ability to accurately quantify hundreds of lipid species in relatively short analysis time [80]. Combined with automated sample handling and data analysis, shotgun lipidomics has garnered favor in high-throughput studies. One of the key features of shotgun lipidomics is that all lipids and matrix components enter the mass spectrometer simultaneously and at a constant rate throughout the acquisition period. As each analyte is subjected to the same ionization conditions, the ion peak intensities of all lipid species within a class are similarly affected. Systematic studies of instrument responses using structurally distinct lipid species have demonstrated that ionization efficiency is predominantly dependent on the lipid polar head group and weakly dependent on the structure of fatty acid moieties [81,82]. Then, the relative ion suppression effect of co-eluting lipid species and classes is supposed to be constant and can therefore be accounted during data analysis. For instance, analysis of equimolar mixture of PC with different acyl chain (with the available molecules) shows no differences in the instruments response once isotopic corrections are applied [83]. However, care must be taken to ensure the sample is not highly concentrated otherwise non-linear ionization response or aggregation effects may be observed. Shogun lipidomic is a great method to obtain the accurate quantification of major lipids in biological sample by MS with a good dynamic range assessed by titrating the amount of a total lipid extract relative to a defined amount of spiked internal lipid standard. However, regarding the international guideline previously discussed, to calculate absolute concentration of lipids, calibration curves and response/correction factors are required [61] which is not available using this technique. Another strategy can consist to use all the available standards to run calibration curves. Absolute quantification is then measured for the corresponding molecules. For other molecules, the calibration curves of the closer’s metabolites are used. This strategy is proposed for human plasma in flow injection analysis (with or without the SelexION) on the Lipydizer platform and allows the accurate quantification of complex lipids [84]. Globally with this shotgun approaches, from a quantitative point of view, we must stay cautious and only talk about accurate quantification.

#### 5.3.2. By Thin Layer Chromatography

Old and efficient classical methods for glycerolipid quantification were based on the separation of the various glycerolipid classes by thin layer chromatography (TLC) followed by their extraction from the TLC and their quantification. It can be carried out by total phosphorous colorimetric quantification which is very efficient but not really sensitive [85], or through the analysis of hydrolyzed fatty acid by GC. Methanolysis of the associated FA to form methyl ester of FA (FAME) is performed. Then, FAMEs are determined and quantified by gas chromatography coupled to flame ionization detector (GC-FID). These methods are robust and have not become obsolete. They allow the absolute quantification of glycerolipids through their FA based on the response of the FID detectors that is linear for a wide range of FA concentrations as well as for a broad range of FA chain lengths [86]. In the same way, a MS detector can be used. Since the main species of fatty acids are commercially available, calibration curves can be performed with a correct dynamic range [87]. However, these methods are time-consuming and not well adapted to high throughput analyses of lipidomes from engineered strains.

#### 5.3.3. By Liquid Chromatography coupled to Mass Spectrometry 

Liquid chromatography coupled to mass spectrometry (LC-MS/MS) is an attractive alternative because of its flow, sensitivity and specificity. As was discussed, the use of internal standards of lipid species is necessary to perform the quantification. Ideally, one ^13^C standard per natural lipid molecule should be used, but these standards are not available for each natural lipid molecule. Thus, one lipid molecule per class of lipids, having two saturated FA, usually not present in the natural extract, is sometimes used as internal standard [88]. However, the use of this internal standard cannot correct biases due to different ionization efficiency of some lipid classes such as galactolipids or phospholipids. That is clearly the limitation when we work without analytical standard for each specie.

To circumvent the discrepancy of ionization efficiency and the absence of available labeled standard for all lipid molecules, we can use a quality control (QC) sample to normalize and correct the quantification. This QC corresponds to a known lipid extract mimicking the studied samples, quantified once by TLC plus GC-FID, and then systematically run with the samples to be analyzed by LC-MS/MS [89]. This method was established and validated for at least five kinds of organisms covering plant, microalgae and yeast: *Arabidopsis thaliana*, *Nannochloropsis sp*., *Phaeodactylum tricornutum*, *Aurantiochytrium limacinum* and *Saccharomyces cerevisiae*. This strategy is based on the glycerolipidome knowledge of the studied samples. For each new studied organism, a large batch of the type of cells to be analyzed is cultivated and its lipid is extracted by an adapted Folch method [89]. The lipid extract represents the QC and is quantified with its FA content by methanolysis and GC-FID. Each lipid class is separated by TLC and quantified by GC-FID or qualified by ion trap mass spectrometry (MS^n^). With these technics, lipid classes are respectively quantified with their FA distribution and lipid molecules are identified with their fatty acid localization on the glycerol backbone as described in [90]. The glycerolipidome of this new organism is then known and MRM transitions can be established for analyzing samples from this organism. To have a better insight of potential biases linked to the nature of FA, the FA profile of each lipid class obtained by TLC plus GC-FID was compared to the profile obtained by MS^n^. The TLC plus GC-FID method allowed a direct access of the FA profile in a given class, whereas the FA profile must be reconstructed with MS technique, knowing the proportion of each molecule constituting this class and the nature of their FA previously established with the MS trap. Profiles obtained with both techniques were found to be comparable, i.e., the major FAs with one method are also the major FAs with the other one [89]. This finding shows no major differences in the estimation of the main molecular species present in each class of glycerolipids. Therefore, there is no huge quantification bias within a class between different lipid molecules bearing at least one unsaturation. These experiments revealed that the defect of quantification is mainly due to the lipid class and to the correcting factor established with its corresponding internal standard. To circumvent this problem, we can use both: internal standards per chromatographic range of elution, to take into account the ionization suppression effect, and an external standard by systematically running in tandem with the unknown sample the QC, previously quantified by TLC plus GC-FID methods. This QC should have a FA composition more or less similar to the samples to be analyzed, usually it corresponds to an extract from the same organism than the samples came from. The QC will determine correction factors for each class of lipid to have a correspondence between the content of the QC measured in TLC +GC-FID and the content measured in LC-MS/MS. These correction factors will be applied for each class of lipid in the sample to obtain an adjusted quantification of lipid molecules. However, because the nature of the FA may also impact the number of ions formed in the ESI source, the absolute quantification of each molecular specie cannot be ascertained.

## 6. Conclusions

Due to the structural complexity of lipids, their quantification is not easy to perform. Different strategies and techniques should be addressed depending on the targeted molecules. Despite the low sensitivity of NMR, it is a quantitative tool and allows the absolute quantification of lipids containing phosphorus by ^31^P NMR, or of simple mixture of lipids by ^1^H NMR, it can permit the absolute quantification of PL, TG, CE, FC or ω-FA for instance. NMR is non-destructive method, then the lipids extract can be analyzed with other techniques afterwards. After a chromatographic separation, the detector used is decisive and can influence the quantification of lipids. With the use of universal detectors such as ELSD or CAD, the detector response is not influenced by the chemical structure (chain length and number of unsaturations) of molecular species in the same lipid class. Therefore, quantification of each lipid class is possible using one standard per concerned class. Concerning mass spectrometry detection method, despite its sensitivity and its practicality, the use of this tool for the quantification of lipids is not without disadvantages. The difference of ionization efficiency between lipid species, depending on their structure, is indeed a real limitation of this technique. The situation is quite convenient for simple lipids because pure standards are available. Thus, calibration curves can be performed to correct detector sensitivity. This tool is then really convenient to propose absolute quantification paying attention to the validation of the method. For complex lipids, the situation is more complicated because pure molecular species are not available for all molecules. Consequently, the lack of calibration curves prevents the absolute quantification of lipids either in shotgun or in flow injection analysis. Thus, absolute quantification for complex lipids should not be established with mass spectrometry detection, we should only propose data in relative quantification (or close/accurate quantification with caution) and respect guidelines to obtain and validate the data (https://lipidomics-standards-initiative.org/guidelines/lipid-species-quantification).

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
