# Peer review of "Quantification of Lipids: Model, Reality, and Compromise"

_biomolecules, 2018, doi:10.3390/biom8040174_

Reviewer 1 Report

The paper presents itself as a review of lipidomics techniques from NMR to MS. It does contain small discussions of these techniques, but with most emphasis on universal detectors and chromatographic methods (sections 4 and 5 in the manuscript). 

I think sections 4 and 5 are valuable for the community, but the NMR section is by no means comprehensive and should either be updated to actually review the field of NMR-based lipidomics or left out. In the present form and extent it doesn't add anything to the field.

Focusing on sections 4 and 5, they list many different studies using slightly different techniques or protocols appropriate for a review, but I miss a summary or a discussion of these different approaches in terms on what they achieved and how this can be used by other researchers. The present manuscript largely remains on the descriptive level, where it would add more to the community if the authors had also taken the discussion to the relational and comparative level.

The paper must be thoroughly corrected for English and grammar errors. In addition, it is full of style inconsistencies, which should be corrected - in the present form it seems rather carelessly produced. Examples are numerous and in some cases so bad that they make it difficult to understand the manuscript:

Line 20: what is meant by "sustainable development of lipidomics"?

Line 21: "analyze and quantify them..." is not a real sentence and should be reworded

Line 22: analyzed should be analyze

Line 23: "State of Art" should not be capitalised and should be "state of the art"

Line 26: The paper uses "quantification" in the rest of the paper. While both quantification and quantitation are used, they are typically not used indiscriminately but consistently either or. See eventually Anal. Chem. 54 (1982) 1456A.

Line 29: No colon after "Introduction"

Line 45: such as

Line 50: NMR as an abbreviation has just been introduced in line 46, so it needs not be explained again.

Line 58: leave out "were" in "several guidelines were resulted from..."

Lines 70-71: Do you mean ".. and only about 80.. are commercially available"?

Lines 72 and 74: "Won't" is slang or spoken language and should be avoided in scientific writing

Line 80: What does their in "...recommended to validate their assays..." refer to - please rephrase.

Line 98: No need to write both NMR and nuclear magnetic resonance

Lines 104-107: Use an inline equation editor. You cannot use a period as a sign of multiplication. Later on (line 195) an asterisk is used, which is not correct either.

Line 114: The term "magnetically equivalent molecular structures" is highly misleading. NMR has the well-defined term "magnetically equivalent spins", which means that the spins provide identical NMR response... But here it is stated that "magnetically equivalent molecular structures" give largely overlapped resonances. The sentence should be rephrased.

Lines 124-125: The sentence should be rephrased for grammar reasons. In addition, it is unclear what the authors want to state with the sentence. The fact that 1H NMR lipidomics is complex does not provide any insight - all the outlined techniques are complex since they involve complex detection or interpretation steps, and all researchers are trained in using complex analyses. So what is the essential point here?

Line 134: 1H should have the number 1 as superscript

Line 135: Importation?

Line 136: FID occurs two times - and is by the way introduced as the abbreviation for "flame ionisation detector" elsewhere...

Line 140 and 145: What are the curly symbols representing?

Line 145: If the nomenclature (1H, s) is important, it should be explained to non-NMR experts that it means one hydrogen and singlet multiplicity.

Line 159: For clarity, please explain why Normal-phase liquid chromatography ends being abbreviated NP-HPLC

Line 163: Please provide references to support the use of SFC in lipid analysis.

Line 164: "Being often free of ..." should be reworded

Line 168: "Conjugated" should not be capitalised, and no need to explain the abbreviation MS again.

Lines 181-184: Please explain the formula by discussing the dependency of the different parameters. Please make sure that the same symbols are used in the formula and in the text below - as it is now, it is impossible to understand. And the paragraph starting in line 182 should probably not be a new paragraph but be part of the paragraph starting line 176.

Section 4.3: This is a listing of applications of the universal detectors, here a discussion and relation of the different studies should also appear.

Line 254: "can be also" -> "can also be"

Line 260: No need to introduce MS again.

Line 268: No need to introduce ESI, which was already introduced in line 168.

Line 285: What is ISTD?

Line 288: Is [Burla...] a reference?

Sections 5.4, 5.5, and 5.6: Be consistent in using colon or not.

Line 390: "Another one strategy propose"... needs rewording and grammar correction

Line 428: Please rephrase the sentence starting "After chromatographic..." so it is clear that it doesn't refer to NMR but is a new technique.

Line 443: The list of abbreviations is not complete. Additionally, for "FA:" delete the colon.

Line 623: "" and "" is not part of the title.

Author Response

The paper presents itself as a review of lipidomics techniques from NMR to MS. It does contain small discussions of these techniques, but with most emphasis on universal detectors and chromatographic methods (sections 4 and 5 in the manuscript). 

I think sections 4 and 5 are valuable for the community, but the NMR section is by no means comprehensive and should either be updated to actually review the field of NMR-based lipidomics or left out. In the present form and extent it doesn't add anything to the field.

For us this NMR part is important to be kept in a manuscript dedicated to the quantification of lipids since this technic provides the possibility to do absolute quantification which is a challenge for this family of molecules. Then we modified this part to make it clearer: NMR section has been updated to be more comprehensive and some references have been added. In this review, we focused on absolute quantification of lipid species, since a recent paper reviewed the field of NMR-based lipidomics in 2017. The objective is to compare the different techniques used in lipidomics, and show the drawbacks and advantages of NMR in the field of absolute quantification of lipid species. In this part, we tried to show the potential of NMR for the quantification of different lipid classes : fatty acids, glycerolipids, phospholipids and lipoproteins.

Focusing on sections 4 and 5, they list many different studies using slightly different techniques or protocols appropriate for a review, but I miss a summary or a discussion of these different approaches in terms on what they achieved and how this can be used by other researchers. The present manuscript largely remains on the descriptive level, where it would add more to the community if the authors had also taken the discussion to the relational and comparative level.

Thanks very much for this comment, we completed the introduction and the conclusion  in this sense. Introduction was completed and a discussion more detailed around the interest and drawbacks of each technique to quantify simple and complex lipids was added in the conclusion.

The paper must be thoroughly corrected for English and grammar errors. In addition, it is full of style inconsistencies, which should be corrected - in the present form it seems rather carelessly produced. Examples are numerous and in some cases so bad that they make it difficult to understand the manuscript:

We thanks the reviewers  to point out all these points and we tried to correct them.

Line 20: what is meant by "sustainable development of lipidomics"?

We thanks the reviewer for this remark , it was not clear the sentence has been changed

Line 21: "analyze and quantify them..." is not a real sentence and should be reworded

The sentence has been reworded

Line 22: analyzed should be analyze

The sentence has been reworded

Line 23: "State of Art" should not be capitalised and should be "state of the art"

We modified this point

Line 26: The paper uses "quantification" in the rest of the paper. While both quantification and quantitation are used, they are typically not used indiscriminately but consistently either or. See eventually Anal. Chem. 54 (1982) 1456A.

We read the note from Anal Chem with interest, and used the term "quantification" in all the paper

Line 29: No colon after "Introduction"

It has been modified

Line 45: such as

It has been modified

Line 50: NMR as an abbreviation has just been introduced in line 46, so it needs not be explained again.

It has been modified

Line 58: leave out "were" in "several guidelines were resulted from..."

It has been modified

Lines 70-71: Do you mean ".. and only about 80.. are commercially available"?

The sentence has been modified

Lines 72 and 74: "Won't" is slang or spoken language and should be avoided in scientific writing

It has been modified

Line 80: What does their in "...recommended to validate their assays..." refer to - please rephrase.

It has been modified

Line 98: No need to write both NMR and nuclear magnetic resonance

Nuclear magnetic resonance has been removed

Lines 104-107: Use an inline equation editor. You cannot use a period as a sign of multiplication. Later on (line 195) an asterisk is used, which is not correct either.

Equation editor was used for the equations.

Line 114: The term "magnetically equivalent molecular structures" is highly misleading. NMR has the well-defined term "magnetically equivalent spins", which means that the spins provide identical NMR response... But here it is stated that "magnetically equivalent molecular structures" give largely overlapped resonances. The sentence should be rephrased.

The sentence has been rephrased

Lines 124-125: The sentence should be rephrased for grammar reasons. In addition, it is unclear what the authors want to state with the sentence. The fact that 1H NMR lipidomics is complex does not provide any insight - all the outlined techniques are complex since they involve complex detection or interpretation steps, and all researchers are trained in using complex analyses. So what is the essential point here?

The essential point here is that the quantification is not easy because of overlapped signals. A paragraph has been added at the beginning of the section b) to explain the complexity of NMR spectra of lipid species.

Line 134: 1H should have the number 1 as superscript

This has been modified

Line 135: Importation?

Importation means data upload. The sentence has been changed.

Line 136: FID occurs two times - and is by the way introduced as the abbreviation for "flame ionisation detector" elsewhere...

In NMR spectroscopy, FID means “Free Induction decay”. The sentence has been changed

Line 140 and 145: What are the curly symbols representing?

Signification of symbols have been indicated. w-9 FA, w-6,7 FA and w-3 FA refer to poly-unsaturated fatty acids.  w-9 FA means “omega-9” fatty acid or n-9 fatty acid. “omega-9” refers to the position of the final double bond in the chemical structure, which is nine carbon atoms from the tail end of the molecular chain.

Line 145: If the nomenclature (1H, s) is important, it should be explained to non-NMR experts that it means one hydrogen and singlet multiplicity.

Thanks for this remark, we found it was not important. The nomenclature has been removed.

Line 159: For clarity, please explain why Normal-phase liquid chromatography ends being abbreviated NP-HPLC

This point has been revised in the text

 Line 163: Please provide references to support the use of SFC in lipid analysis.

It has been added

Line 164: "Being often free of ..." should be reworded

This has been modified

Line 168: "Conjugated" should not be capitalised, and no need to explain the abbreviation MS again.

This has been modified

Lines 181-184: Please explain the formula by discussing the dependency of the different parameters. Please make sure that the same symbols are used in the formula and in the text below - as it is now, it is impossible to understand. And the paragraph starting in line 182 should probably not be a new paragraph but be part of the paragraph starting line 176.

Thank you for your comment. The formula is discussed in the new version of the manuscript explaining the influence of the different parameters on the response and on the particle size. In addition, the symbols used in the formula and in the text have been corrected and homogenized.

Section 4.3: This is a listing of applications of the universal detectors, here a discussion and relation of the different studies should also appear.

As recommended in your remark, a discussion, which addresses the relation of the different studies about the use of universal detectors as well as other points, now appears in this section.

Line 254: "can be also" -> "can also be"

This has been modified

Line 260: No need to introduce MS again.

This has been modified

Line 268: No need to introduce ESI, which was already introduced in line 168.

This has been modified

Line 285: What is ISTD?

It means internal standard  : this has been added in the text and in the list of abbreviation

Line 288: Is [Burla...] a reference?

It has been added

Sections 5.4, 5.5, and 5.6: Be consistent in using colon or not.

This has been modified

Line 390: "Another one strategy propose"... needs rewording and grammar correction

This has been modified

Line 428: Please rephrase the sentence starting "After chromatographic..." so it is clear that it doesn't refer to NMR but is a new technique.

It has been rephrased

Line 443: The list of abbreviations is not complete. Additionally, for "FA:" delete the colon.

This has been modified

Line 623: "" and "" is not part of the title.

This has been modified

Reviewer 2 Report

This review is well consolidated but some references are not appropriately selected. Thus, authors should cite appropriate reference. In addition, as far as I know, Lipidyzer is shotgun lipidomics with ion mobility selection and not liquid chromatography lipidomics.

Author Response

This review is well consolidated but some references are not appropriately selected. Thus, authors should cite appropriate reference. In addition, as far as I know, Lipidyzer is shotgun lipidomics with ion mobility selection and not liquid chromatography lipidomics.

Thanks very much for this nice comment. You are completely right, we apologized for the confusion we made, the Lipidyzer part were moved to the "shotgun paragraph".

Round  2

Reviewer 1 Report

The authors have improved the scientific parts of the manuscript according to my recommendations, and the aim of the paper is clearer now. Also many of the grammar and formatting errors have been corrected, but the manuscript still could be improved, especially there remain quite a few formatting issues in the revised document, for example inconsistent use of a multiplication sign (should be ·) but appears both as x (sign of a cross product) and *...